# FEATHERS: Federated Architecture and Hyperparameter Search

## Abstract

Deep neural architectures have profound impact on achieved performance in many of today's AI tasks, yet, their design still heavily relies on human prior knowledge and experience. Neural architecture search (NAS) together with hyperparameter optimization (HO) helps to reduce this dependence. However, state of the art NAS and HO rapidly become infeasible with increasing amount of data being stored in a distributed fashion, typically violating data privacy regulations such as GDPR and CCPA. As a remedy, we introduce FEATHERS—**FE**derated **A**rchi**T**ecture and **H**yp**ER**parameter **S**earch, a method that not only optimizes both neural architectures and optimization-related hyperparameters jointly in distributed data settings, but further adheres to data privacy through the use of differential privacy (DP). We show that FEATHERS efficiently optimizes architectural and optimization-related hyperparameters alike, while demonstrating convergence on classification tasks at no detriment to model performance when complying with privacy constraints.

## 1 Introduction

Federated learning (FL) is a distributed machine learning paradigm aiming to learn a shared model on data distributed at different locations without ever exchanging the data itself (McMahan et al., 2017). It is a promising solution in several industries, such as finance or healthcare, where it is infeasible to share the data due to privacy and security regulations. As in classical machine learning (ML), neural architectures and optimization-related hyperparameters (hence simply referred to as hyperparameters) have to be selected in FL before training. Since even experts are likely to choose non-optimal architectures and hyperparameters, different neural architecture search (NAS) and hyperparameter optimization (HO) methods have been developed to automatically search for suitable architectures/hyperparameters (Kairouz et al., 2021; Zoph & Le, 2017; Pham et al., 2018; Liu et al., 2019; Agrawal et al., 2021). With HO- and NAS-methods experts only have to define a search space over candidates instead of defining a specific rigid architecture and setting hyperparameters for a given ML-task. A search strategy is then applied to automatically find the optimal element within this space.

To date, most NAS- and HO-methods are designed for classical ML-settings. As more and more data is being stored decentralized and privacy awareness is rising (He et al., 2020a; Khodak et al., 2021), a number of approaches to perform NAS/HO in FL settings have lately been proposed. However, the latter still face a significant number of challenges. First, current methods either optimize neural architectures *or* hyperparameters; a serious obstacle as performing NAS and HO sequentially is costly, especially in FL settings where it is preferable to minimize the communication performed between devices. In addition, the choice of architectures and hyperparameters inherently depend on each other. For instance, adopting a deeper architecture may require selecting different learning rates in order to assure adequate update scaling, following gradient back-propagation through the network. Therefore architectures and hyperparameters should be optimized jointly. A second major challenge is that NAS- and HO-methods are traditionally not designed to be privacy-preserving. Throughout FL training, the server and clients exchange the updated parameters several times. In light of the growing concern about the disclosure of personal information(Fredrikson et al., 2015) and the threat of adversarial attacks (Ye et al., 2022), both ML models and their training process, and hence NAS and HO approaches, should guarantee privacy in distributed settings. To address the above

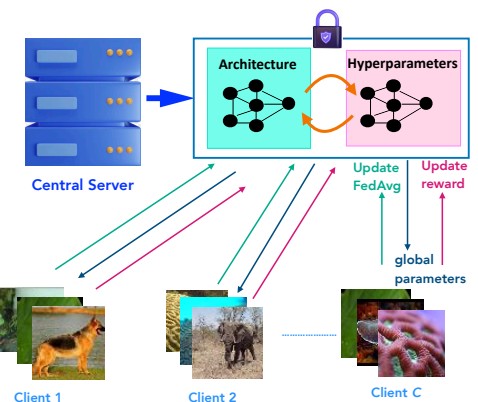

Figure 1: **FEATHERS Overview.** FEATH-ERS jointly optimizes neural architectures and hyperparameters using data distributed across clients $\mathcal{C}$ while providing privacy-guarantee.

Table 1: **FEATHERS unifies NAS, HO and DP in federated learning settings**. FEATHERS is the first method that jointly optimizes neural architectures (NAS) and arbitrary other hyperparameters (HO) in a federated learning (Fed.) setting while providing privacy guarantees (DP). Existing SOTA methods do not tackle all of these challenges in a unified way.

| Method | NAS | HO | DP | Fed. |
|---|---|---|---|---|
| DARTS(Liu et al., 2019) | ✓ | ✗ | ✗ | ✗ |
| DP-FNAS(Singh et al., 2020) | ✓ | ✗ | ✓ | ✓ |
| FedNAS(He et al., 2020a) | ✓ | ✗ | ✗ | ✓ |
| DP-FTS-DE(Dai et al., 2021) | ✗ | ✓ | ✓ | ✓ |
| FedEx(Khodak et al., 2021) | ✗ | ✓ | ✗ | ✓ |
| **FEATHERS** | ✓ | ✓ | ✓ | ✓ |

challenges, we propose a novel method: **FEATHERS**[1] - **FE**derated **A**rchi**T**ecture and **H**yp**ER**parameter **S**earch. As illustrated in Table 1, FEATHERS is the first method to synergize architecture search and hyperparameter optimization, while enabling privacy preserving federated learning.

The overall architecture of FEATHERS is shown in Fig. 1. In essence, it consists of an HO- and a NAS-phase executed in an alternating fashion. Motivated by the bandit based HO of Khodak et al. (2021), a $n$-bandit game is played to identify promising hyperparameters for a subsequent NAS-phase. Then, a NAS-phase is performed for multiple iterations using the identified hyperparameters. Differentiable, cell-based, NAS (Liu et al., 2019) is used to allow "averaging" architectures of several clients using FedAvg, thus obtaining a new global architecture. The global architecture obtained is then sent back to the clients and used for further training. The differentiability of the architecture entails a crucial property of FEATHERS: enabling a privacy preserving optimization-scheme using differential privacy (DP).

Overall, we make the following contributions:

1. We propose a novel method, FEATHERS, that jointly optimizes neural architectures and hyperparameters in distributed data settings.

2. We prove that the HO phase of FEATHERS converges with high probability and that FEATHERS' convergence properties subsequently coincide with those of DARTS if the HO phase converges.

3. By exploiting differentiability of model- and architectural parameters we provide privacy-guarantees during the search- and evaluation-stage via DP.

4. We empirically show that FEATHERS converges towards well-suited architectures and hyperparameters on various classification tasks.

We proceed as follows: After presenting related work we formally introduce the problem before moving to our proposed solution. A convergence analysis is then conducted, followed by our experimental evaluation of FEATHERS on several classification tasks. Finally, we conclude our work and show possible directions for future work.

## 2 Related Work

**Hyperparameter Optimization.** Several works address hyperparameter optimization in federated learning (Koskela & Honkela, 2018; Mostafa, 2019). Genetic CFL(Agrawal et al., 2021) clusters edge devices based

---

[1]We make our code publicly available at: `https://anonymous.4open.science/r/FEATHERS-250B/`.

on the training hyperparameters and genetically modifies the parameters clusterwise. DP-FTS-DE(Dai et al., 2021) integrates differential privacy into federated Thompson sampling with distributed exploration to preserve privacy and uses it for federated hyperparameter tuning. FLoRA(Zhou et al., 2021) uses a single-shot task by querying each client for several instantiations once and selects the best instantiations based on what the clients returned w.r.t. learning progress. FedEx(Khodak et al., 2021) uses a weight-sharing mechanism for hyperparameter optimization in the federated setting. In contrast, FEATHERS follows a few-shot policy as we adjust the hyperparameters several times during training.

**Neural Architecture Search.** NAS aims to automatically identify an optimal neural architecture for a given task. Most methods are based on reinforcement learning (RL) (Zoph & Le, 2017; Zoph et al., 2016; 2018), evolutionary algorithms (EAs) (Xie & Yuille, 2017; Galván & Mooney, 2021; Darwish et al., 2020) or gradient decent (GD) (Liu et al., 2019; Dong & Yang, 2019; Xie et al., 2018; Li et al., 2020a). Gradient based methods have been found to be more robust compared to the former (Zhu et al., 2021) and thus we adapt gradient-based neural architecture search in FEATHERS. Since this approach is differentiable it can be used in federated setting. Some recent NAS-methods for FL include Fed-NAS (He et al., 2020a) that uses the gradient-based NAS method MiLeNAS (He et al., 2020b) for personalized federated learning, and DP-FNAS (Singh et al., 2020), which also adopted differentiable NAS (Liu et al., 2019) combined with DP.

**Differential Privacy** DP has first been introduced in (Dwork, 2006) to protect private information in a dataset from queries based on arbitrary mechanisms. In recent years, select works have shown that ML-models, especially neural networks, carry private information of their training data in their parameters (Fredrikson et al., 2015). To protect such private data from leaking, DP has been successfully employed in various FL settings. In particular, it has been shown that SGD can be turned into a differentially private algorithm by simply adding an appropriate amount of noise to the parameters during training (Abadi et al., 2016).

## 3 FEATHERS

Our objective is to efficiently optimize neural architectures and hyperparameters in a joint manner in FL under privacy guarantees. We now define our problem setup formally and present our proposed solution.

### 3.1 Problem Definition

We consider a federated learning setting with a set of clients $\mathcal{C}$ of size $C$, each holding a dataset $D_1, ..., D_C$. The data of each client $c$ is split into training $\langle \mathbf{X}_{train}^{(c)}, \mathbf{y}_{train}^{(c)} \rangle$ and validation data $\langle \mathbf{X}_{val}^{(c)}, \mathbf{y}_{val}^{(c)} \rangle$ that is used to solve a supervised learning task. We aim to find an architecture $\mathbf{a} \in \mathcal{A}$ and hyperparameters $\mathbf{h} \in \mathcal{H}$ minimizing the global validation loss over all clients. Formally we phrase the problem as follows:

$$\min_{\mathbf{a},\mathbf{h}} \sum_{c \in \mathcal{C}} v_c \cdot \mathcal{L}_{\mathbf{a},\mathbf{h}}(\mathbf{w}^*, \mathbf{X}_{val}^{(c)}, \mathbf{y}_{val}^{(c)}) \quad \text{with} \tag{1}$$

$$\mathbf{w}^* = \arg\min_{\mathbf{w}} \sum_{c \in \mathcal{C}} v_c \cdot \mathcal{L}_{\mathbf{a},\mathbf{h}}(\mathbf{w}, \mathbf{X}_{train}^{(c)}, \mathbf{y}_{train}^{(c)}) \tag{2}$$

where $\mathbf{w} \in \mathbb{R}^n$ refers to the parameters of a neural network (model parameters) and $v_c$ is the weight of a client $c$:

$$v_c := \frac{|\mathbf{X}_{val}^{(c)}|}{\sum_{c \in \mathcal{C}} |\mathbf{X}_{val}^{(c)}|} \tag{3}$$

Note that in a FL settings, the global validation loss is a weighted sum of the client's local validation losses. From now on, we denote the global training- and validation loss as $\mathcal{L}_{\mathbf{a},\mathbf{h}}(\mathbf{w}, \mathbf{X}_{train}, \mathbf{y}_{train})$ and $\mathcal{L}_{\mathbf{a},\mathbf{h}}(\mathbf{w}, \mathbf{X}_{val}, \mathbf{y}_{val})$ respectively. Additionally, we require our method to be $\epsilon$-differential w.r.t. model parameters, architectural parameters and rewards. For better readability, we omit DP in the definition of the optimization problem and will return to it in Section 3.3.

## 3.2 FEATHERS Architecture

FEATHERS operates in two stages, the *search stage* and the *evaluation stage*. The search stage consists of an alternating procedure: As a first step, an instantiation of hyperparameters is identified which is expected to achieve a high decrease in validation loss. In a second step, the identified instantiation is used to perform several optimization-steps of the architecture. The two steps are repeated until convergence.

In the evaluation stage the optimized architecture is retrained. Again, the HO-scheme from the search stage is applied to optimize hyperparameters. We now describe the HO and NAS phase in detail.

**Hyperparameter Optimization (HO).** To identify well-working hyperparameters $\mathbf{h}$ we have to solve the following objective:

$$\mathbf{h}^* = \arg\min_{\mathbf{h}} \mathcal{L}_{\mathbf{a}^*, \mathbf{h}}(\mathbf{w}^*; \mathbf{X}_{val}, \mathbf{y}_{val}) \tag{4}$$

Here, $\mathbf{w}^*$ denote model parameters minimizing the training-loss under architecture $\mathbf{a}^*$ and $\mathbf{a}^*$ are the architectural parameters minimizing the validation-loss under hyperparameters $\mathbf{h} \in \mathcal{H}$ where $\mathcal{H}$ is a discrete set of hyperparameter instantiations. We solve the above by using a $n$-armed bandit-approach with a strategy similar to $\epsilon$-greedy as shown in Algorithm 1. **(Line 1-3)**: We start of by intializing the parameters, architecture and reward estimates. **(Line 4-13)**: We then randomly sample $m$ hyperparameter instantiations from a distribution $\pi$ over $\mathcal{H}$. For each sampled instantiation one communication round of training is performed using the same weights $\mathbf{w}$ and architecture $\mathbf{a}$. This yields an approximation of $\mathbf{a}^*$ and $\mathbf{w}^*$, denoted as $\mathbf{w}'$ and $\mathbf{a}'$ respectively. Each client computes its local validation loss before and after performing local training using hyperparameters $\mathbf{h}$ as shown in Algorithm 2 **(Line 1-5)**. The loss before local training is denoted as $\ell_{\mathbf{a}, \mathbf{w}}^{(c)}$ and the loss after local training as $\ell_{\mathbf{a}', \mathbf{w}'}^{(c)}$. We compute the reward-signal $r_{\mathbf{h}}^{(e)}$ indicating how well instantiation $\mathbf{h}$ performed in HO-phase $e$ as:

$$r_{\mathbf{h}}^{(e)} = \sum_{c \in \mathcal{C}} v_c \cdot \left( \ell_{\mathbf{a}, \mathbf{w}}^{(c)} - \ell_{\mathbf{a}', \mathbf{w}'}^{(c)} \right) \tag{5}$$

In the above equation $v_c$ refers to the weight of each client. After testing each sampled $\mathbf{h}$ we obtain a vector $\mathbf{r}^{(e)}$ where each entry corresponds to one hyperparameter instantiations in $\mathcal{H}$: For instantiations $\mathbf{h}$ sampled in HO-round $e$, $\mathbf{r}^{(e)}$ contains the reward, all other entries are zero. The reward-estimates $\mathbf{r}$ are then updated using $\mathbf{r}^{(e)}$ by applying the update rule:

$$\mathbf{r} = \mathbf{r} + (\mathbf{i} \circ \alpha \circ (\mathbf{r}^{(e)} - \mathbf{r})) + ((1 - \mathbf{i}) \circ (\alpha \circ \mathbf{r} - \mathbf{r})) \tag{6}$$

Here, $\circ$ is the Hadamard product, $\mathbf{i}$ is a binary vector indicating which hyperparameter instantiations were sampled in exploration round $e$ and $\alpha$ is a constant factor determining how aggressively the reward-estimate should be updated. If a hyperparameter instantiation was sampled at round $e$, its reward in $\mathbf{r}$ will be corrected by the error of the current reward estimate. All instantiations that were not sampled in $e$ get scaled down by $\alpha$ since well suited instantiations in an early stage of training might not be suitable in later stages anymore. For example, in the beginning of training, an instantiation with a higher learning rate might be more appropriate whereas in later training-stages lower learning rates should be chosen.

The use of the reward estimates is three-fold: (1) The hyperparameter instantiation with the highest reward achieved so far will be used to train the supernet in the next NAS-phase. (2) Reward estimates determine the number of communication-rounds in the HO-phase before the HO-phase is starting: For this, we first compute the distribution $\pi = \text{softmax}(\mathbf{r})$ over instantiations using the reward estimates $\mathbf{r}$. This allows to compute the entropy $H$:

$$H = \sum_{\mathbf{h} \in \mathcal{H}} \ln(\pi(\mathbf{h})) \cdot \pi(\mathbf{h}) \tag{7}$$

The number of HO-rounds performed next is determined by $\kappa = \text{rnd}(\beta H)$. Here, $\beta$ is a parameter to control the exploration-exploitation trade-off. In the beginning, all rewards are set to 0, thus leading to a uniform distribution which has the maximum entropy. Over time the reward-estimates reflect which instantiations work well and which do not. Hence, $\pi$ gets less uniform and the entropy decreases over time, favoring exploitation over exploration in later training stages. (3) The distribution $\pi$ is used to sample hyperparameter-instantiations tested in the next HO-round.

**Algorithm 1:** FEATHERS method server side

**Data:** set of clients $\mathcal{C}$, client weight $v_c \forall c \in \mathcal{C}$
**Data:** hyperparameter search space $\mathcal{H}$
**Data:** architecture search space $\mathcal{A}$

1   initialize parameters $\mathbf{w}$;
2   initialize architecture $\mathbf{a}$;
3   initialize reward estimates $\mathbf{r} \leftarrow \mathbf{0}$;
4   $\pi \leftarrow \text{softmax}(\mathbf{r})$;
5   **for** $p$ *in phases* **do**
6     **if** $p == $ *'ho'* **then**
7       sample $n$ hyperparameters $\mathbf{h}$ from $\pi$;
8       $\mathbf{r}_p \leftarrow \mathbf{0}$;
9       **for** $h$ *in* $\mathbf{h}$ **do**
10         $l_1, l_2, \mathbf{w}^*, \mathbf{a}^* \leftarrow \text{client\_step}(h, \mathbf{w}, \mathbf{a})$;
11         $\mathbf{r}_p[h] \leftarrow \sum_{c \in \mathcal{C}} v_c \cdot (l_1^{(c)} - l_c^{(c)})$;
12       $\mathbf{r} \leftarrow \text{update\_rewards}(\mathbf{r}, \mathbf{r}_p)$;
13       $\pi \leftarrow \text{softmax}(\mathbf{r})$;
14     **if** $p == $ *'nas'* **then**
15       $\mathbf{h}^* \leftarrow \mathcal{H}[\arg\max_{\mathbf{h}} \mathbf{r}[\mathbf{h}]]$;
16       **for** $j$ *in* $nas\_steps$ **do**
17         $\mathbf{w}, \mathbf{a} \leftarrow \text{client\_steps}(\mathbf{h}^*, \mathbf{w}, \mathbf{a})$;

**Algorithm 2:** FEATHERS Framework Client-side Search stage

**Data:** Network parameters $\mathbf{w}$, architecture $\mathbf{a}$
**Data:** Hyperparameter configuration $\mathbf{h}$
**Data:** Data $\mathbf{X}_{train}, \mathbf{X}_{val}, \mathbf{y}_{train} \mathbf{y}_{val}$

1   $l_1 \leftarrow \mathcal{L}_{\mathbf{a},h}(\mathbf{w}, \mathbf{X}_{val}, \mathbf{y}_{val})$;
2   $\mathbf{w}^* \leftarrow SGD(\nabla_{\mathbf{w}} \mathcal{L}_{\mathbf{a},h}(\mathbf{w}, \mathbf{X}_{train}, \mathbf{y}_{train}), \mathbf{w}, \mathbf{h})$;
3   $\mathbf{a} \leftarrow SGD(\nabla_{\mathbf{a}} \mathcal{L}_{\mathbf{a},h}(\mathbf{w}^*, \mathbf{X}_{val}, \mathbf{y}_{val}), \mathbf{a}, \mathbf{h})$;
4   $\mathbf{w} \leftarrow SGD(\nabla_{\mathbf{w}} \mathcal{L}_{\mathbf{a},h}(\mathbf{w}, \mathbf{X}_{train}, \mathbf{y}_{train}), \mathbf{w}, \mathbf{h})$;
5   $l_2 \leftarrow \mathcal{L}_{\mathbf{a},h}(\mathbf{w}, \mathbf{X}_{val}, \mathbf{y}_{val})$;
6   **return** $l_1, l_2, \mathbf{w}, \mathbf{a}$;

**Algorithm 3:** FEATHERS Framework Client-side Evaluation stage

**Data:** Network parameters $\mathbf{w}$, architecture $\mathbf{a}$
**Data:** Hyperparameter configuration $\mathbf{h}$
**Data:** Data $\mathbf{X}_{train}, \mathbf{X}_{val}, \mathbf{y}_{train} \mathbf{y}_{val}$

1   $l_1 \leftarrow \mathcal{L}_{\mathbf{a},h}(\mathbf{w}, \mathbf{X}_{val}, \mathbf{y}_{val})$;
2   $\mathbf{w} \leftarrow SGD(\nabla_{\mathbf{w}} \mathcal{L}_{\mathbf{a},h}(\mathbf{w}, \mathbf{X}_{train}, \mathbf{y}_{train}), \mathbf{w}, \mathbf{h})$;
3   $l_2 \leftarrow \mathcal{L}_{\mathbf{a},h}(\mathbf{w}, \mathbf{X}_{val}, \mathbf{y}_{val})$;
4   **return** $l_1, l_2, \mathbf{w}, \mathbf{a}$;

**Neural Architecture Search.** Once the HO-phase yields a hyperparameter instantiation $\mathbf{h}$, the architecture is optimized under $\mathbf{h}$ for a certain number of communication rounds as shown in Algorithm 1 **(Line 14-17)**, thereby solving:

$$\min_{\mathbf{a}} \mathcal{L}_{\mathbf{a},\mathbf{h}}(\mathbf{w}^*, \mathbf{X}_{val}, \mathbf{y}_{val}) \tag{8}$$

$$\text{where } \mathbf{w}^* = \arg\min_{\mathbf{w}} \mathcal{L}_{\mathbf{a},\mathbf{h}}(\mathbf{w}, \mathbf{X}_{train}, \mathbf{y}_{train}) \tag{9}$$

Inspired by Differentiable Architecture Search (DARTS) (Liu et al., 2019) we solve this optimization problem as follows: We define our search space to be a space over *cells*. A cell is a Directed Acyclic Graph (DAG) in which each node is a feature representation and each edge is a *mixed operation*. The feature representation of some node $z$ is computed using all its parent-nodes and the mixed operations defining the edges between $z$ and its parent, i.e. for some node $z_j$ the representation is computed as: $z_j = \sum_{i<j} o^{(z_i, z_j)}(\mathbf{x}^{z_i})$ Here, $o^{(z_i, z_j)}$ is a mixed operation and $\mathbf{x}^{z_i}$ is the feature representation of node $z_i$. A mixed operation connecting nodes $z_1$ and $z_2$ is defined as a weighted sum over a set of operations $\mathcal{O}$:

$$o^{(z_1, z_2)} = \sum_{o \in \mathcal{O}} \frac{\exp(a_o^{(z_1, z_2)})}{\sum_{o' \in \mathcal{O}} \exp(a_{o'}^{(z_1, z_2)})} o(\mathbf{x}) \tag{10}$$

Here, $a_o^{(z_i, z_j)}$ are the architectural parameters to be learned. Since they fully describe the architecture, we will refer to them as the architecture $\mathbf{a}$ from now on. Objective 8 is solved by an alternating optimization of the architecture and model parameters. First, the architecture is updated by following the gradient $\nabla_{\mathbf{a}} \mathcal{L}_{\mathbf{a},\mathbf{h}}(\hat{\mathbf{w}}, \mathbf{X}_{val}, \mathbf{y}_{val})$ where $\hat{\mathbf{w}} = \mathbf{w} - \eta \nabla_{\mathbf{w}} \mathcal{L}_{\mathbf{a},\mathbf{h}}(\mathbf{w}, \mathbf{X}_{train}, \mathbf{y}_{train})$. As a second step the model parameters are updated by following the gradient $\nabla_{\mathbf{w}} \mathcal{L}_{\mathbf{a},\mathbf{h}}(\mathbf{w}, \mathbf{X}_{train}, \mathbf{y}_{train})$. As shown in Algorithm 2 **(Line 2-4)**, parameter-updates are performed on client-side in each communication round and yield new architectural and model parameters $\mathbf{a}'_c$ and $\mathbf{w}'_c$ for each client $c$ respectively. Since both, the architectural and model parameters, are parameters of a non-convex optimization problem with a differentiable loss-function, we use FedAvg to aggregate the model- and architecture parameters of all clients after each communication round. We use two types of cells: Normal cells and reduction cells. Normal cells keep the dimensions of the input while reduction cells apply an additional reduction-operation.

**Discretizing the Architecture.** Since differentiable NAS requires a continuous relaxation of the architectural sapce $\mathcal{A}$, the architecture learned by FEATHERS has to be discretized after training. This is done

**Algorithm 4:** FEATHERS Framework Client-side Search stage with DP

---

**Data:** Parameters $\mathbf{w}$ and architecture $\mathbf{a}$
**Data:** Hyperparameter configuration $h$
**Data:** Data $\mathbf{X}_{train}$, $\mathbf{X}_{val}$, $\mathbf{y}_{train}$ $\mathbf{y}_{val}$
1 $l_1 \leftarrow \mathcal{L}_{\mathbf{a},h}(\mathbf{w}, \mathbf{X}_{val}, \mathbf{y}_{val}) + N_{l_1}$;
2 $\mathbf{w}^* \leftarrow SGD(\nabla_{\mathbf{w}}\mathcal{L}_{\mathbf{a},h}(\cdot), \mathbf{w}, \mathbf{h})$;
3 $\mathbf{a} \leftarrow SGD(\frac{1}{B}\sum_{i=1}^{B}\nabla_{\mathbf{a}}\mathcal{L}_{\mathbf{a}}(\cdot) + N_i, \mathbf{a}, \mathbf{h})$;
4 $\mathbf{w} \leftarrow SGD(\frac{1}{B}\sum_{i=1}^{B}\nabla_{\mathbf{w}}\mathcal{L}_{\mathbf{a}}(\cdot) + N_i, \mathbf{w}, \mathbf{h})$;
5 $l_2 \leftarrow \mathcal{L}_{\mathbf{a},h}(\mathbf{w}, \mathbf{X}_{val}, \mathbf{y}_{val}) + N_{l_2}$;
6 **return** $l_1$, $l_2$, $\mathbf{w}$, $\mathbf{a}$;

**Algorithm 5:** FEATHERS Framework Client-side Evaluation stage with DP

---

**Data:** Parameters $\mathbf{w}$ and architecture $\mathbf{a}$
**Data:** Hyperparameter configuration $h$
**Data:** Data $\mathbf{X}_{train}$, $\mathbf{X}_{val}$, $\mathbf{y}_{train}$ $\mathbf{y}_{val}$
1 $l_1 \leftarrow \mathcal{L}_{\mathbf{a},h}(\mathbf{w}, \mathbf{X}_{val}, \mathbf{y}_{val}) + N_{l_1}$;
2 $\mathbf{w} \leftarrow SGD(\frac{1}{B}\sum_{i=1}^{B}\nabla_{\mathbf{w}}\mathcal{L}_{\mathbf{a}}(\cdot) + N_i)$;
3 $l_2 \leftarrow \mathcal{L}_{\mathbf{a},h}(\mathbf{w}, \mathbf{X}_{val}, \mathbf{y}_{val}) + N_{l_2}$;
4 **return** $l_1$, $l_2$, $\mathbf{w}$, $\mathbf{a}$;

by selecting the top $k$ operations with the highest architectural weight over all cells. Also, no operation is allowed to connect the same two nodes. The discretized architecture is then retrained in the evaluation stage where only the HO phase from Algorithm 1 is applied as discussed above. Each client performs standard gradient descent w.r.t. the model parameters as shown in Algorithm 3 **(Line 1-3)**.

## 3.3 Differential Privacy

Although in FL no data is exchanged between server and clients, the parameters sent to the server still leak private information (Fredrikson et al., 2015). It has been shown that differential privacy (DP) can be used to provably protect private information encoded in these parameters during Stochastic Gradient Descent (SGD) (Abadi et al., 2016). We adapt this notion to both, model parameters and architectural parameters since both inherently depend on the data the model is trained on. DP was introduced in Dwork (2006) and is defined as follows:

**Definition 1.** *For any two datasets $D$, $D'$ that differ in exactly one record a mechanism $M$ is called $\epsilon$-differential private if $\forall x : Pr[M(D) = x] \leq \exp(\epsilon)Pr[M(D') = x]$ holds where $Pr[M(D) = x]$ refers to the probability of mechanism $M$ outputting $x$ if executed on $D$.*

In our case $M$ is the learning procedure, i.e. SGD. Making SGD differential private can be achieved by clipping gradients and adding Gaussian noise to the gradient of each sample w.r.t. the parameters, resulting in an algorithm called DP-SGD (Abadi et al., 2016). Updating model- or architectural parameters with DP-guarantees then becomes:

$$\theta \leftarrow \alpha_\theta \frac{1}{B} \sum_{i=1}^{B} \nabla_\theta \mathcal{L}_{\mathbf{a}}(\mathbf{w}, \mathbf{x}^{(i)}) + \mathcal{N}(0, \sigma_\theta^2 C_\theta^2 \mathbf{I}) \tag{11}$$

In the above equation we use $\theta \in \{\mathbf{w}, \mathbf{a}\}$ to either refer to the model- or architectural parameters. $B$ denotes the batch-size, $\mathcal{N}$ is the normal distribution, $\sigma_\theta$ is a scaling-parameter, $C_\theta$ is the maximum gradient norm, $\mathbf{I}$ the identity matrix and $\alpha_\theta$ is the learning rate for parameters $\theta$. We use DP-SGD for learning both, the model parameters and the architecture. Hence, our method enjoys all convergence- and privacy-guarantees given by DP-SGD which can be controlled via the parameter $\epsilon$ (Abadi et al., 2016). As $\epsilon$ inversely depends on noise-parameters $\sigma$, for high $\epsilon$-values DP-SGD achieves approximately SGD-convergence while losing privacy-guarantees. For low $\epsilon$ we obtain strong privacy guarantees while losing convergence-guarantees. It should be noted that FedAvg averages the parameters that have been computed by the clients. Since DP is closed under arbitrary post-processing, averaging does not break DP (Dwork et al., 2014). Similarly, we apply DP on losses sent to the server to "hide" possible private information from data by adding small Gaussian noise with zero mean to the losses. Algorithm 4 shows the DP variant of the search stage of FEATHERS clients. It performs the exact same computations as Algorithm 2 except that it adds independent noise $N_i$ to the gradient of each sample $i$ **(Line 2-4)**. Also, independent Gaussian noise is added to the losses computed on client side which are used to compute the rewards for each hyperparameter configuration **(Line 1, 5)**. The noise is drawn from a Gaussian distribution with zero mean and variance depending on the privacy budget $\epsilon$ (lower $\epsilon$ means higher variance). Algorithm 5 shows the DP variant of the evaluation stage and performs the same computation as Algorithm 3. Again, the only difference is that the DP variant adds independent

Gaussian noise $N_i$ with zero mean and variance depending on $\epsilon$ to each sample $i$ and independent Gaussian noise to the losses computed on client side.

## 3.4 Convergence Analysis

We now show that FEATHERS' convergence properties in distributed settings coincide with the convergence properties of DARTS in centralized settings with high probability, only scaled by a controllable factor arising from using FedAvg. For simplicity we do not consider adding DP in our analysis.

**Theorem 1.** *Given a joint distribution $p(X_1, \ldots, X_n, y)$ over random variables $X_1, \ldots, X_n, y$ from which each client $c \in \mathcal{C}$ of a set of clients $\mathcal{C}$ samples a dataset $\langle \mathbf{X}^{(c)}, \mathbf{y}^{(c)} \rangle \sim p$, FEATHERS enjoys the same convergence properties as DARTS in a centralized setting if applied on a dataset $\langle \mathbf{X}, \mathbf{y} \rangle$ where $\mathbf{X} = \bigcup_{c \in \mathcal{C}} \mathbf{X}^{(c)}$ and $\mathbf{y} = \bigcup_{c \in \mathcal{C}} \mathbf{y}^{(c)}$.*

*Proof.* We treat the HO-phase of FEATHERS as an oracle and assume that it returns optimal hyperparameters $\mathbf{h}^*$. Once $\mathbf{h}^*$ was obtained, it is fixed for a certain number of communication rounds $\kappa$. In each communication round $i$ epochs of DARTS are performed locally on each client. Since we employ FedAvg to average model parameters after $i$ local epochs, we exploit that FedAvg converges with rate $\mathcal{O}(\frac{1}{i\kappa})$ (Li et al., 2020b). Since FedAvg converges and parameter-updates are only propagated during NAS-phases, it follows that FEATHERS enjoys the same convergence properties as DARTS in each NAS-phase scaled by the convergence of FedAvg $\mathcal{O}(\frac{1}{i\kappa})$. $\qquad\square$

Since the above proof assumes that our method selects optimal hyperparameters $\mathbf{h}^*$ for each NAS-phase, we will now show that the HO-phase converges with high probability in non-stationary bandit-environments.

**Theorem 2.** *Given a fixed hyperparameter-space $\mathcal{H}$ and noisy, non-stationary rewards $r_{\mathbf{h}}^{(j)} \sim \mathcal{N}(\mu_{\mathbf{h}}^{(j)}, \sigma_{\mathbf{h}})$ where $\mu_{\mathbf{h}}^{(j)}$ is the expected value of the reward at iteration $j$, $\sigma_{\mathbf{h}}$ its standard deviation and $\mathbf{h} \in \mathcal{H}$, the HO-strategy of FEATHERS is at most off by $\alpha \cdot 3\sigma_{\mathbf{h}}$ for learning rate $\alpha$ with probability $0.997$ once $\mathbf{h} \in \mathcal{H}$ is sampled.*

*Proof.* Our proof is inspired by convergence results for $\epsilon$-greedy strategies as stated in (Sutton & Barto, 2018). We assume that $|\mu_{\mathbf{h}}^{(j+1)} - \mu_{\mathbf{h}}^{(j)}| \le \delta$ for finite $\delta \in \mathbb{R}$ in all iterations and $0 < \alpha < 1$ in the update rule. Since the softmax-function cannot evaluate to a point-mass, we can make a strict positivity assumption of the distribution over hyperparameters, i.e. $\pi_i[\mathbf{h}] > 0$ for all $\mathbf{h} \in \mathcal{H}$. Thus, with $j$ approaching infinity, each $\mathbf{h} \in \mathcal{H}$ will be sampled infinitely many times. At an iteration $j$, in the most extreme case, a certain $\mathbf{h} \in \mathcal{H}$ has not been sampled yet. Assume it gets sampled in iteration $j$. Since $r_{\mathbf{h}}^{(j)} \sim \mathcal{N}(\mu_{\mathbf{h}}^{(j)}, \sigma_{\mathbf{h}})$ and the current estimate reward-estimate $\mathbf{r}_{\mathbf{h}} = 0$, the update rule reads: $\mathbf{r}_{\mathbf{h}} = \alpha \cdot r_{\mathbf{h}}^{(j)}$. Since we assume all rewards being Gaussian distributed, the probability of obtaining a reward $r_{\mathbf{h}}^{(j)}$ in the range of $3\sigma_{\mathbf{h}}$ is $0.997$. Since $0 < \alpha < 1$ holds, our estimate is at most $\pm\alpha \cdot 3\sigma_{\mathbf{h}}$ of w.r.t. $\mu_{\mathbf{h}}^{(j)}$ in 99.7% of the cases. $\qquad\square$

As the above only considers the case in which our algorithm terminates after some $\mathbf{h} \in \mathcal{H}$ is sampled, we also have to consider the following case: Assume $\mathbf{h}$ is sampled at iteration $j$ and a reward-estimate is obtained. After that, $\mathbf{h}$ is not sampled for $k$ subsequent iterations. The following theorem gives bounds for how much off our estimate will be in this case.

**Theorem 3.** *Under the assumptions of Theorem 2, the reward estimate $r_{\mathbf{h}}^{(j+k)}$ will be at most off by $\alpha^k r_{\mathbf{h}}^{(j)} - (k\delta + \mu_{\mathbf{h}}^{(j)})$ assuming that $\mathbf{h}$ is sampled at iteration $j$ and not sampled for $k$ subsequent iterations.*

*Proof.* By assumptions from Theorem 2, the mean will be shifted by at most $k\delta$ after $k$ steps. Since the update rule for $r_{\mathbf{h}}$ is defined as $r_{\mathbf{h}} = \alpha r_{\mathbf{h}}$, the reward estimate after $k$ iterations in which $\mathbf{h}$ is not sampled is $\alpha^k r_{\mathbf{h}}^{(j)}$. It follows that, $k$ iterations after $\mathbf{h}$ was sampled, the reward estimate is off by at most $\alpha^k r_{\mathbf{h}}^{(j)} - (k\delta + \mu_{\mathbf{h}}^{(j)})$. $\qquad\square$

It turns out that the above bound can be controlled by setting $\alpha \le (1 + \frac{k\delta}{\mu^{(j)}})^{\frac{1}{k}}$ assuming we have access to $\mu^{(j)}$ (see Appendix F). In the case $\mu^{(j+1)} - \mu^{(j)} = \delta$, this relation guarantees that our reward estimate of

some $\mathbf{h}$ is still optimal if $\mathbf{h}$ was not sampled for $k$ HO-rounds. Since we can assume that the loss decreases between HO-rounds, i.e. $\mu^{(j+1)} - \mu^{(j)} < 0$, the assumption $0 < \alpha < 1$ used in the above theorems is not violated. Using Theorem 2 we can assume that we have an estimate of $\mu^{(j)}$ fulfilling at least $\mu^{(j)} \pm \alpha \cdot 3\sigma_{\mathbf{h}}$ with high probability for some $\mathbf{h}$ sampled the first time in round $j$. Hence, the errors of reward-estimates can be controlled within reasonable bound given by Theorem 3 in subsequent rounds.

## 4 Experiments and Results

In order to empirically demonstrate that FEATHERS is capable of jointly optimizing neural architectures and hyperparameters in FL settings with privacy guarantees, we investigate the following three questions:

**Q1.** Can FEATHERS compete with state of the art HO- and NAS-methods in FL settings at various scales and label skews?

**Q2.** Does FEATHERS adjust the choices of instantiations over time to account for dynamics of training process?

**Q3.** How well does FEATHERS perform if DP is employed to preserve privacy with respect to privacy-budget $\epsilon$?

We next describe our experiment protocol including the employed datasets before presenting our results in detail.

### 4.1 Experimental Protocol

In our experiments, we analyzed FEATHERS on three image classification tasks: Fashion-MNIST which contains black-white images of 10 different articles of clothing to be categorized as well as CIFAR-10 and Tiny-Imagenet which contain colored images of 10/200 different categories respectively. The fourth task is a binary classification problem on a fraud detection dataset which contains anonymized bank-account data from bank-customers based on which the fraud risk (high or low) has to be predicted (see Appendix A). All datasets were partitioned randomly on a set of clients such that each client holds the approximately same number of samples. Since in FL it is common to have data unequally distributed across clients, we also conducted experiments in which we introduced a label skew in the data (referred to as *ls* subsequently) .We first executed the search stage of FEATHERS using a search space over CNN/MLP-architectures. For the evaluation-stage we used the best normal cell and reduction cell obtained in the search stage to build up a larger network (validation networks). For discretizing the architecture $k = 2$ was chosen in order to be comparable to other cell-based NAS-methods. We then retrained the validation network and optimized hyperparameters using the same HO-strategy as in the search stage. The results of the validation network were then reported. Since we assume a cross-silo setting, we allowed all clients to participate in each communication round. Additionally we tested both FEATHERS with and without DP for fraud detection, to show that adding DP does not prevent learning a suitable architecture. Appendix B contains a detailed description of our search space.

We implemented FEATHERS in Python based on the flower framework for federated learning. All models were built using PyTorch and the clients were distributed on Nvidia DGX-clusters with A-100 40GB GPUs. Also the server was deployed on the same cluster, however, using a separate GPU to simulate a cross-silo federated learning setting with parallel client-execution.

### 4.2 Results

We are now ready to answer the posed research questions and will elaborate on each of them in more detail.

**(Q1) FEATHERS achieves SOTA, independently of scale and label skew.** First, we show that FEATHERS, while performing both HO and NAS, achieves state of the art results on Fashion-MNIST, CIFAR-10 and Tiny-Imagenet in Table 2. Despite DARTS being based on hyperparameters that have traditionally manually been tuned for best results by humans, our method beats DARTS (94% vs. 92% on

Table 2: **Achieved accuracies of FEATHERS and baselines.** DARTS, FedEx and FEATHERS were compared in different federated learning settings as described in Section 4. Each experiment was performed 5 times and the mean accuracy and standard deviation are reported. Colors are interpolated from green to blue (high accuracy to low accuracy).

| Dataset | Fashion-MNIST | | CIFAR-10 | | Tiny-Imagenet | |
|---|---|---|---|---|---|---|
| | w/o ls | w/ ls | w/o ls | w/ ls | w/o ls | w/ ls |
| DARTS (f, 5 clients)[†] | $0.92 \pm 0.02$ | $0.93 \pm 0.01$ | $0.92 \pm 0.02$ | $0.90 \pm 0.02$ | $0.67 \pm 0.02$ | $0.67 \pm 0.02$ |
| DARTS (f, 10 clients)[†] | $0.91 \pm 0.02$ | $0.92 \pm 0.02$ | $0.91 \pm 0.03$ | $0.89 \pm 0.04$ | $0.68 \pm 0.02$ | $0.67 \pm 0.03$ |
| DARTS (f, 100 clients)[†] | $0.92 \pm 0.03$ | $0.91 \pm 0.03$ | $0.91 \pm 0.02$ | $0.89 \pm 0.03$ | $0.67 \pm 0.02$ | $0.67 \pm 0.03$ |
| FedEx (5 clients)* | $0.82 \pm 0.01$ | $0.81 \pm 0.01$ | $0.53 \pm 0.02$ | $0.54 \pm 0.03$ | $0.43 \pm 0.04$ | $0.41 \pm 0.04$ |
| FedEx (10 clients)* | $0.78 \pm 0.03$ | $0.78 \pm 0.02$ | $0.51 \pm 0.04$ | $0.51 \pm 0.03$ | $0.41 \pm 0.03$ | $0.40 \pm 0.04$ |
| FedEx (100 clients)* | $0.65 \pm 0.03$ | $0.64 \pm 0.04$ | $0.46 \pm 0.05$ | $0.47 \pm 0.04$ | $0.38 \pm 0.04$ | $0.38 \pm 0.05$ |
| FEATHERS (5 clients) | $0.94 \pm 0.01$ | $0.93 \pm 0.02$ | $0.93 \pm 0.03$ | $0.91 \pm 0.03$ | $0.69 \pm 0.02$ | $0.69 \pm 0.03$ |
| FEATHERS (10 clients) | $0.93 \pm 0.01$ | $0.93 \pm 0.03$ | $0.92 \pm 0.02$ | $0.89 \pm 0.04$ | $0.68 \pm 0.02$ | $0.68 \pm 0.02$ |
| FEATHERS (100 clients) | $0.94 \pm 0.02$ | $0.93 \pm 0.03$ | $0.90 \pm 0.03$ | $0.89 \pm 0.03$ | $0.68 \pm 0.03$ | $0.67 \pm 0.03$ |

*Training performed using architecture found by DARTS.
†The same hyperparameter-settings as described in (Liu et al., 2019) were used.

Table 3: **GPU days comparison.** FEATHERS' runtime is approximately 0.4 GPU-days higher than the runtime of DARTS. FedEx has lower runtimes compared to both, FEATHERS and DARTS, mainly because it does not optimize the architecture and performs less exploration than FEATHERS. All runtimes in GPU-days.

| | Fashion-MNIST | CIFAR-10 | Tiny-Imagenet | Fraud Detection |
|---|---|---|---|---|
| FedEx (5 Clients) | 0.8 | 0.9 | 1.3 | 0.09 |
| FedEx (10 Clients) | 0.8 | 0.8 | 1.3 | 0.07 |
| FedEx (100 Clients) | 0.6 | 0.7 | 1.1 | 0.05 |
| DARTS (5 Clients) | 1.8 | 2.1 | 3.1 | 0.4 |
| DARTS (10 Clients) | 1.7 | 2.0 | 2.9 | 0.3 |
| DARTS (100 Clients) | 1.5 | 1.8 | 2.7 | 0.2 |
| FEATHERS (5 Clients) | 2.2 | 2.5 | 3.6 | 0.6 |
| FEATHERS (10 Clients) | 2.1 | 2.4 | 3.5 | 0.6 |
| FEATHERS (100 Clients) | 1.9 | 2.1 | 3.1 | 0.35 |

Fashion-MNIST, 93% vs. 92% on CIFAR-10, 69% vs. 67% on Tiny-Imagenet) in most distributed learning settings while optimizing for a larger set of parameters. The same holds for label-skew scenarios. Introducing label skews does not seem to adversely affect its performance. However, if label-skew is present, a slight increase in the variance of our results can be seen.

Second, to assess scalability of FEATHERS, we consider variance in results for increasing number of clients (marked in the rows of Table 2). We find that, in contrast to FedEx, the number of participating clients does not seem to have a negative influence on the performance of FEATHERS. We hypothesize that FEATHER's stability is due to more extensive exploration. For an increasing number clients, each client holds a smaller subset of data since the datasets used have fixed size. Thus the stochastic gradients per client tend to have a higher variance, which in turn leads to higher variances in parameter-changes across clients. FEATHERS tests a set of hyperparameter instantiations on a frozen model before applying them for parameter-updates, whereas FedEx directly applies the chosen instantiations. Consequently, higher variance of gradients and FedEx' higher risk of choosing inappropriate hyperparameters can lead to poorly performing models. In contrast, FEATHERS tends to choose "safer" instantiations.

Finally, in terms of runtime, FEATHERS ($\sim 2.5$ GPU-days) does not add significant overhead compared to DARTS ($\sim 2$ GPU-days). The additional HO-phase during the search stage adds an overhead of approximately 0.1-0.8 GPU-days, depending on the number of instantiations tested in each HO-round. In contrast, FedEx' runtime ($\sim 1$ GPU-day) is much lower compared to DARTS and FEATHERS since FedEx does not perform

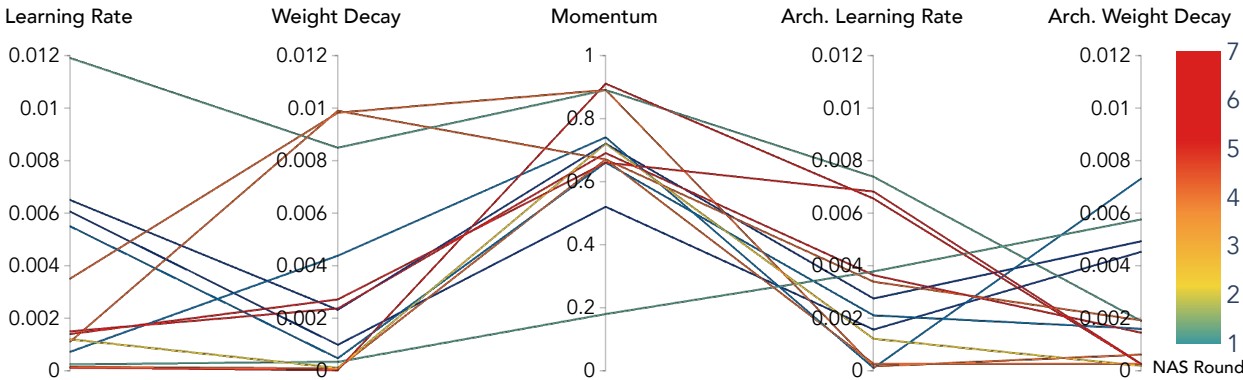

Figure 2: **FEATHERS adjusts hyperparameters over time.** The choices of hyperparameters are adapted during training to optimize the validation loss. In earlier stages (blue lines) higher learning rates are chosen whereas in later stages of training (red lines) lower learning rates are chosen. The figure shows hyperparameter-selections of three FEATHERS-runs on CIFAR-10. (Best viewed in color)

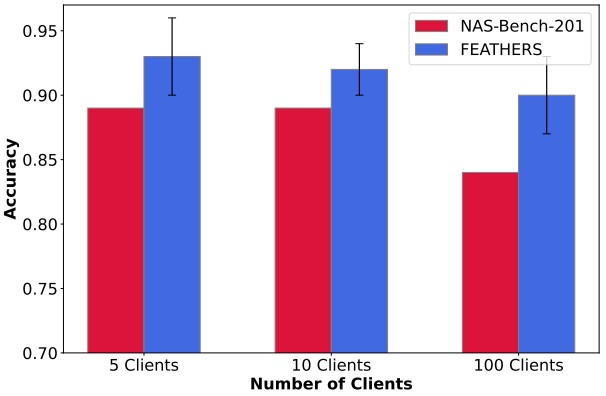

Figure 3: **FEATHERS beats NAS-Bench-201**. FEATHERS' additional HO phase improves the performance of a fixed architecture. Note that NAS-Bench-201 only provides averaged results in centralized settings.

Figure 4: **DP retains performance.** For $\epsilon \geq 1$, FEATHERS achieves nearly the same performance with DP as without DP regardless of the number of clients. For low $\epsilon$-values (i.e. stronger privacy-guarantees), the performance decreases.

NAS and that it performs less exploration than our method. See Table 3 for a detailed listing of runtimes w.r.t. datasets and number of clients.

**(Q2) FEATHERS dynamically adjusts hyperparameters.** Figure 2 shows the hyperparameters selected by FEATHERS over time for three runs on CIFAR-10. We observe that our method chooses more "cautious" hyperparameters than engineers usually do. For example, in DARTS it is common to start with a learning rate of 0.025, FEATHERS however chooses much lower learning rates most of the time. Presumably this is due to the properties of our HO-algorithm: In the first HO-round it samples and tests a small subset of instantiations from $\mathcal{H}$ before greedily selecting the one leading to the highest decrease in validation loss. In this concrete example, this choice might lead FEATHERS to choose a lower learning rate than 0.025, simply because there was no better sample. In subsequent HO-rounds the goal is to learn a distribution over instantiations maximizing the reward in the long run. As SGD never truly converges due to its inherent stochasticity, a smaller learning rate is ultimately beneficial in the later stages of training, in order to avoid heavily perturbing away from a minimum (i.e. too large learning rates will "overshoot").

Consequently, FEATHER's "cautious" instantiations entail more stable convergence. In that sense, FEATH-ERS mimics an annealing mechanism in later training stages, which find frequent use in Deep Learning problems. To assess the effect of dynamic hyperparameter adjustments, we compare FEATHERS with NAS-Bench-201 (Dong & Yang, 2020). This benchmark provides a database which allows to query the performance of architectures trained under fixed and manually tuned hyperparameters. The architectures in NAS-Bench-201 were chosen such that they cover widely used architecture search space, including ours. We can easily assess whether our additional HO mechanism helps improving model performance by comparing to NAS-Bench-201: We first run the search stage of FEATHERS to optimize the architecture for 5/10/100 clients. Note that the architecture found can vary for a different number of clients. Then, we train the architecture found during the search stage using FEATHER's validation stage (i.e. with adjustments of hyperparameters) and compare the accuracy of the same architecture reported in NAS-Bench-201 (i.e. trained with fixed hyperparameters) on CIFAR-10. Figure 3 demonstrates that our dynamic adjustment helps improving model performance. This observation further supports our claim that our method adjusts hyperparameters appropriately over time.

**(Q3) FEATHERS preserves privacy.** To demonstrate that FEATHERS provides privacy guarantees without sacrificing predictive performance, we performed classification on the fraud detection dataset. The hyperparameter search space and the architecture search space are described in Appendix B. The same privacy budget $\epsilon \in \{0.2, 0.25, 0.5, 1.0, 1.5, 2.0, \infty\}$ was used for DP applied to the losses, model parameters and architecture parameters. It is noteworthy that a privacy budget of $\infty$ corresponds to FEATHERS without DP. The search stage was performed for 100 communication rounds, all other parameters were set as above. Note that the dataset is heavily skewed (95% negative class, 5% positive class), we thus report F1-scores instead of accuracy. We further used oversampling of positive samples on the client-side to account for label-skew. Figure 4 visualizes the results for different privacy budgets $\epsilon$. For $\epsilon \geq 1$ we obtained a F1-score of approximately 0.77. This means, FEATHERS-DP performs equally well as FEATHERS as long as $\epsilon$ is chosen larger to be larger than 1. Decreasing $\epsilon$ adds more noise on the gradients which increases the privacy level while disturbing the gradient-signal. For $\epsilon \leq 0.5$ we obtained a significant decrease of the F1-score. This confirms that the gradients carrying less (private) information also get less useful for parameter-updates.

In summary, FEATHERS-DP retains the performance of FEATHERS for appropriate $\epsilon$. Finally, we emphasize that adding DP came with approximately 1.5-2 times longer runtimes on our setup. A reasonable trade-off to accommodate privacy considerations. The underlying reason is that for DP the gradient of each sample has to be manipulated, resulting in poorer parallel execution of automatic differentiation. Hence, two trade-offs have to be considered when using DP: (1) Finding a good balance between a high performing model and privacy guarantees and (2) determining whether a longer runtime for training is still practically feasible to protect privacy.

## 5 Conclusion

We have introduced FEATHERS, a federated learning method that efficiently optimizes both neural architectures and hyperparameters jointly, while preserving privacy of the underlying training-data. Our empirical investigation demonstrates that FEATHERS is more than competitive with state of the art NAS-algorithms, despite popular approaches like DARTS only performing a subset of the above, while also optimizing in a larger space of hyperparameters.

FEATHERS now allows for a myriad of real-world problems to be addressed fully, e.g. tasks surrounding finance, defence, or healthcare. As one example, hospitals can use it to collaboratively train cancer-detection models without sharing sensible patient-data and without revealing the patient's identity through parameters. That said, we note that fully automating critical systems could be risky and a human in-the-loop should evaluate the quality of the model found before it is deployed and monitor its prediction.

From a technical perspective, it is further desirable to relax FEATHER's requirement of discrete hyperparameter search spaces in order to properly account for continuous hyperparameters in future work. Incorporating a state reflecting the current state of training in HO-phases to allow for more informed hyperparameter choices instead of a stateless bandit approach is a promising additional direction. Lastly, considering FL-scenarios

other than the cross-silo set-up considered in this work is intriguing, also with respect to enabling above mentioned applications.

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
