# OpenReview forum: "FEATHERS: Federated Architecture and Hyperparameter Search"
_TMLR — Rejected by TMLR_

### Review · Reviewer_R8pX · 2023-07-24

**Summary Of Contributions:**

The paper focuses on a federated learning setup where multiple clients are training a model collaboratively. The problem that the authors focus on is how multiple clients could not only train a model, but do the neural architecture search and hyper parameter optimization for it together at the same time. The authors propose a two stage approach, the first is search and second evaluate, and the search has an alternating approach, where there are two phases at each round, one is the hyper parameter optimization and second is neural architecture search using DARTS. The hyper parameter phase consists of sampling a vector of hyper parameters and evaluating a reward on them, on the validation set. The neural arch search step optimizes the architecture parameters and the model parameters at the same time. The authors also show a version of their proposal which incorporates DP-SGD in its process (with the federated averaging process), to make it differentially private.

**Audience:**

Yes

**Broader Impact Concerns:**

I think a short discussion of fairness in FL, and how different clients/users would get different gains from it would be interesting, as in sometimes users with fewer samples get more benefit, or on the contrary, underrepresented users don't get much benefit.

**Claims And Evidence:**

Yes

**Requested Changes:**

My main concerns with this paper are about the differentially private training and accountings. in more details:

1. when algorithm 2 is changed to algorithm 4 to be DP (and 3 to 5), lines 1 and 2 do not seem to have noise addition/clipping, so there is basically nothing DP there, which confuses me, since both those steps look at client data and they extract information that relies on private data so I think there is leakage happening there? I can see that part of it is on val data, but its client side so I assume the client val data is also private? also line 2 is on train data so its not clear to me why its not DP.  Same goes for line 1 of algorithm 5, and also line 3, I am not sure why the loss is not calculated in a DP manner, as it is a metric from training data. if val data is considered non-private then this is not a concern (it still is for all 4 line 2)

2. The authors do not really explain how they do the accounting. As in for the hyper parameter search, in algorithm 1, there is the for loop for every element in the vector h, is that accounted for in the DP? I believe the added iterations would waste a lot of privacy budget so the results in figure 4 surprise me, as in they are too good, given all the extra added iterations of hparam search.

3. The data is partitioned to train and val for user basis, how is that accounted for in the subsampling for DP accounting?

**Strengths And Weaknesses:**

Strengths:

mixing hyper parameter search with architecture search is interesting and relevant, especially collaboratively, and the fact that the authors are considering privacy too is very important. The authors do a good job of baking all this into the process.

Weaknesses:

My biggest concern with this paper is the DP part, the way the accounting is done and the empirical results. I will elaborate on this the requested changes, but the summary is I am not sure how the authors are doing the privacy accounting, especially with respect to the multiple rounds of evaluation/training steps for the Hyperparam search and reward evaluation.

---

> ### Author Response · Authors · 2023-08-21
> **Rebuttal**
>
> Thank you reviewer R8pX for your concrete critique and pointers of improvement.
>
> We hope to answer the comments in the following:
>
> > No DP in Algorithm 4 and 5
>
> Thank you for pointing out that this point is unclear! We agree that in line 1 and line 5 (Alg. 4) and line 1 and 3 (Alg. 5) no DP is applied in the pseudocode. We corrected this by adding a noise term. The experimental results remain the same as we applied DP on the losses in our experiments. Regarding line 2 in Alg. 4, note that $\mathbf{w}^*$ is only used locally by each client, therefore it is not necessary to apply DP here since $\mathbf{w}^*$ is not shared with the server.
>
> > The authors do not really explain how they do the accounting.
>
> In our experiments we applied the same DP mechanism on the architecture- and network parameters as well as on the losses $l_1$ and $l_2$ sent back to the server (i.e. same privacy budget $\epsilon$ was used). We made this more clear in the algorithms (see your first requested change) as well as in the experiments. Since the same $\epsilon$ is used for all parameters/losses sent back to the server, we don't think that the results in Fig. 4 are surprising. Setting $\epsilon$ to higher values results in a lower standard deviation of the noise distribution, hence the parameters/losses sent to the server are closer to their original values which allows for more stable training and better results.
> Also note that we only iterate over a small sub-sample of hyperparameter configurations, depending on the entropy of the distribution over the hyperparameters. Typically we test ~10-15 configurations.
>
> > The data is partitioned to train and val for user basis, how is that accounted for in the subsampling for DP accounting?
>
> As we understand your question, you ask for how we ensure that the client's training **and** validation set is kept private with DP. As you correctly identified, we are using both, the training data and the validation data during the search stage. The validation data, however, is only used to update the model architecture since we aim to obtain an architecture which allows for good generalization. Since we apply DP to both, the model- and architectural parameters, we ensure that both, the training data and validation data is kept private. Also, we apply DP on the loss values sent to the server to ensure that there is no data leakage in our HPO component. Please elaborate further on this if something is unclear. Thanks!
>
> > Discussion of fairness in FL
>
> Thanks for sharing your thoughts! We agree that fairness in FL is an important and interesting topic, also in the light of HPO and NAS. However, our contribution aims to propose a method to optimize hyperparameters and neural architectures in FL with privacy guarantees, hence we think that a discussion on fairness in FL would be orthogonal here since we do not provide any insights/experiments regarding fairness. Could you please elaborate a bit more why you would find a discussion on fairness appropriate/necessary for our paper? Thank you!

---

### Review · Reviewer_fB8p · 2023-08-05

**Summary Of Contributions:**

The paper proposes a framework to jointly learn neural architecture and hyperparameters in the federated learning setting. Specifically, in every round of federated learning,  it formalizes the problem into a bi-level minimization problem. The outer minimization aims to search for the best hyperparameter given the optimal architecture and its weights, using the multi-bandit search. After achieving the optimal hyperparameters, it uses the continuous differential neural architecture search method to find the optimal weights and architecture given the searched hyperparameters. The algorithm goes iteratively until convergences. Some theoretical analyses have been done to show the proposed method could converge and the experimental results on several datasets show the proposed method could consistently outperform the previous baselines.

**Audience:**

Yes

**Broader Impact Concerns:**

I don't see any ethical concerns.

**Claims And Evidence:**

Yes

**Requested Changes:**

Edits on the paper presentations:
1.  All current table titles are vague and need explicitly mention the metric used. For example, Table 2 should be "the achieved accuracy with variants". Table 3 should be "the GPU days comparison".
2. I suggest changing Figure 2's x-axis to be the iterations. It would better present the changes across the training than the current form.
3. Figure 3 shows the proposed methods in the NAS-bench 201. However, it is not clear which baselines are used. I believe it is the DARTS and it should be explicitly mentioned.
Experiments:
1.  The neural architecture search part only considers DARTS. As DARTS have been shown not that reliable in several works [1,2],  It would be better to show other search methods as well.


[1] Zela, Arber, et al. "Understanding and Robustifying Differentiable Architecture Search." International Conference on Learning Representations. 2019.
[2] Wang, Ruochen, et al. "Rethinking Architecture Selection in Differentiable NAS." International Conference on Learning Representations. 2020.

**Strengths And Weaknesses:**

Pros:
1. The proposed problem is important in building a better-federated learning framework. The paper claims it is the first work to do the hyperparameter and neural architecture search jointly while keeping the differential privacy guarantee.
2. The paper is well-written and easy to follow.
3. The experiments show the proposed method could outperform the previous baselines in terms of both accuracy and efficiency.

Cons:
1. The theoretical analysis is not strict and gives limited information.
2. The paper does the hyperparameter search and neural architecture search alternatively, which is also widely used in other domains than federated learning settings. In other words, this framework is simply adding the procedure into the federated learning setting.
3. The experiments were only done in the DARTS algorithm and it may cause some problems since DARTS is not stable in a lot of search spaces.



Minor:
Typos: P8 bottom 0.69% vs 0.67% -> 69% vs 67%

---

> ### Author Response · Authors · 2023-08-21
> **Rebuttal**
>
> Thank you reviewer fB8p for the detailled review. It really helped us in improving our paper to its new version.
>
> In the following we wish to provide our comments on each of our changes and questions / suggestions:
>
> > The theoretical analysis is not strict and gives limited information.
>
> We agree that our theoretical analysis does not provide full information on convergence. However, the focus of our paper is the methodology and empirical evaluation. Our theoretical analysis is only supposed to show that (1) we achieve the same convergence as DARTS in the NAS phase and (2) our bandit based approach converges.
>
> > The paper does the hyperparameter search and neural architecture search alternatively [...]
>
> We assume that you mean "in an alternating fashion" instead of "alternatively". In that sense we agree that such schemes are used in other domains. We think applying such a scheme in a federated setting and providing extensive empirical evaluations is already an important contribution for the field since it shows that these schemes can be successfully applied in FL as well.
>
> > All current table titles are vague and need explicitly mention the metric used
>
> We changed the table titles and now mention the metrics being used explicitly/clearer. Thank you for pointing this out.
>
> > I suggest changing Figure 2's x-axis to be the iterations.
>
> The reason why we decided to use a parallel plot was that the scale of the different parameters under consideration should not be important. For instance, the momentum's range is [0, 1] while the learning rate's range is [0, 15]. Putting this into a figure where the x-axis represents the iterations, it would be hard to see the changes for parameters with a small scale such as the learning rate.
>
> > Figure 3 shows the proposed methods in the NAS-bench 201. However, it is not clear which baselines are used. I believe it is the DARTS and it should be explicitly mentioned.
>
> Figure 3 should demonstrate the the HO phase of FEATHERS helps improving the performance of a fixed architecture. To this end we ran the search stage of FEATHERS to identify an architecture. We then queried NAS-Bench-201 for the architecture's results when it is trained with a fixed set of hyperparameters. Also, we ran the evaluation stage of FEATHERS (which performs hyperparameter optimization) and compared the results. We made this more clear in the figure's caption and in the text.
>
> > The neural architecture search part only considers DARTS. As DARTS have been shown not that reliable in several works [1,2], It would be better to show other search methods as well.
>
> We applied FEATHERS with a more robust version of DARTS by introducing path dropout (PDrop) which has to be shown to effectively mitigate architecture degeneration in _Understanding and Robustifying Differentiable Architecture Search. Zela et al. 2020._ See some preliminary results on CIFAR-10 in the table below. We are running further experiments on FashionMNIST and Tiny-Imagenet and will add it to our results.
>
> |      |    no label skew  |   label skew   |
> | ---- | ---- | ---- |
> |   FEATHERS + PDrop (5 Clients)   |   0.92 $\pm$ 0.02   |   0.91 $\pm$ 0.02   |
> |   FEATHERS + PDrop (10 Clients)  |   0.92 $\pm$ 0.02   |   0.90 $\pm$ 0.04   |
> |   FEATHERS + PDrop (100 Clients)  |   0.90 $\pm$ 0.03  |   0.90 $\pm$ 0.03   |

---

### Review · Reviewer_ocob · 2023-08-08

**Summary Of Contributions:**

The paper presents a framework for automating hyperparameter optimization and neural architecture search in a federated setting. Additionally, the paper illustrates how to integrate differential privacy to ensure privacy constraints. Hyperparameters and architecture are optimized in a round-robin manner, based on the validation data gathered from all clients. The proposed method, named FEATHER, improves upon DARTS - a popular neural architecture search method from the literature - on image classification benchmarks.


**Audience:**

Yes

**Broader Impact Concerns:**

I do not expect the paper to have any ethical implication. However, if such a system would possibly deployed it might provide some privacy preserving guarantees due to the use of differential privacy.

**Claims And Evidence:**

Yes

**Requested Changes:**


## Questions:

- What are the convergence properties of the DARTS algorithm?
- Equation 11 appears to be unclear. While it seems that parameters and architecture are updated together, examining Equations 8 and 9 suggests that they are updated alternately in accordance with the DARTS approach.
- What's the rationale behind confining the optimization to discrete hyperparameters?
- Why is a single round of training deemed a suitable approximation for optimal parameters w and architecture parameters a?
- In Equation 5, why is the difference taken here rather than the most recent observed validation error? What does w'/a' represent?
- Why was the choice made not to jointly optimize hyperparameters and architectural parameters? Consider, for instance:

JAHS-Bench-201: A Foundation For Research On Joint Architecture And Hyperparameter Search
Archit Bansal, Danny Stoll, Maciej Janowski, Arber Zela, Frank Hutter


- In the experiments, why is the skewed label experiments referred to as non-i.i.d.? How exactly are data points sampled?
- How does FedEx work?

## Typos:

- cell-baded -> cell-based
- for a multiple iterations ->  for multiple iterations
- discreete -> discrete



**Strengths And Weaknesses:**

## Strength

- To the best of my knowledge, this is the first paper that combines differential privacy and hyperparameter optimization / neural architecture search.



## Weaknesses

I found the paper to be confusingly written. In particular, the methodology section is difficult to understand. I also lack an understanding of the motivation behind separating hyperparameter and architecture optimization instead of treating it as a joint optimization problem. Moreover, several papers (e.g Yang et al.) have highlighted that DARTS doesn't outperform randomly sampling architectures. The paper fails to explain the rationale for choosing DARTS over more recent approaches.

I am missing some motivation why and how differential privacy (DP) extends to architectural parameters? If so, what drives this decision? Is there an anticipation of sensitive information being embedded in the architecture?

Lastly, I also have some concerns regarding the empirical results outlined in the paper. Firstly, the outcomes between DARTS and FEATHERS don't appear to exhibit significant differences, considering the overlapping uncertainty bars. Additionally, I find the absence of an ablation study concerning the impact of hyperparameter optimization and neural architecture search on the overall performance. Furthermore, the experimental section states that FEATHERS surpasses state-of-the-art HPO methods. Could the paper clarify the specific baseline that is considered as the state-of-the-art reference?

NAS evaluation is frustratingly hard
Antoine Yang, Pedro M. Esperança, Fabio M. Carlucci

---

> ### Author Response · Authors · 2023-08-21
> **Rebuttal (1/2)**
>
> Thank you reviewer ocob for the detailed review and for all the pointers you've provided that helped improving our paper.
>
> Below we'd like to elaborate on each of the points of your review in more detail:
>
> > I also lack an understanding of the motivation behind separating hyperparameter and architecture optimization instead of treating it as a joint optimization problem.
>
> We don't separate the hyperparameter and architecture optimization. Rather we view it as another bi-level optimization problem (similar to DARTS where optimizing the architecture is viewed as a bi-level optimization problem). We'd like to clarify our method briefly: **Search Stage:** Our method first searches for an approximation of a good/the optimal hyperparameter configuration and then performs NAS for several iterations under the configuration identified. This process is repeated until convergence, i.e. until a good architecture is found. **Evaluation Stage:** Once a good architecture is found, we retrain the entire network. During the training we apply our hyperparameter optimization scheme again, i.e. we first optimize hyperparameters for the neural architecture search and then optimize hyperparameters of the training algorithm training the architecture identified (e.g. SGD).
>
> We adapted some parts of our methodology part to be clearer.
>
> > Moreover, several papers (e.g Yang et al.) have highlighted that DARTS doesn't outperform randomly sampling architectures. The paper fails to explain the rationale for choosing DARTS over more recent approaches.
>
> We are aware of these works. To the best of our knowledge this is mainly due to the fact that DARTS yields degenerate architectures in a wide range of search spaces. Works like [1] aim to deal with these problems and can be easily incorporated in FEATHERS as well. We already did this and performed some experiments with the more robust variant of DARTS and added the results.
>
> > I am missing some motivation why and how differential privacy (DP) extends to architectural parameters?
>
> Architectural parameters as used in DARTS can be interepreted as a probability that a certain operation defined in the search space is optimal for minimizing the loss. This implies - assuming DARTS yields the correct "distribution" over operations - an attacker can easily infer which operation(s) transform the data best s.t. the loss is optimized. Hence architectural parameters can guide attackers regarding "where to search" for private information in the network. This can be mitigated/avoided by applying DP on architecture parameters as well.
>
> > Firstly, the outcomes between DARTS and FEATHERS don't appear to exhibit significant differences, considering the overlapping uncertainty bars
>
> We agree that there are overlaps in the uncertainty bars. However, we emphasize that we solve a harder optimization problem than DARTS does since we optimize for additional hyperparamters. Nevertheless we still achieve better results than DARTS on average.
>
> > Additionally, I find the absence of an ablation study concerning the impact of hyperparameter optimization and neural architecture search on the overall performance
>
> We do have an ablation study regarding the impact of optimizing hyperparamters (see Figure 3). Also, we show the impact of the neural architecture search since FedEx applies a similar hyperparameter optimization strategy as we do but with a fixed architecture (see Table 2).
>
> > Could the paper clarify the specific baseline that is considered as the state-of-the-art reference?
>
> We consider FedEx to be one state of the art method we compare to (see Table 2).
>
> > What are the convergence properties of the DARTS algorithm?
>
> We do not provide exact convergence results of DARTS. To the best of our knowledge there is no rigor convergence analysis on DARTS. Since our focus lies in the methodology and our theoretical analysis of FEATHERS is supposed to show that (1) we inherit convergence guarantees of DARTS and (2) our hyperparameter optimization method converges sufficiently well, we did not elaborate more on this.
>
> > Equation 11 appears to be unclear.
>
> We summarized the DP version of Eq. 8 and Eq. 9 in Eq. 11 by generalizing it to multiple parameters that can be optimized for (architecture **or** model parameters). We intended to save space with this since the equation would be the same for architecture and model parameters. We hope that it's clear now, we added a line clarifying this point in the paper.

---

> > ### Comment · Reviewer_ocob · 2023-08-25
> > **response to rebuttal**
> >
> > First of all, I thank the authors for responding to my review and updating the paper.
> >
> > > We are aware of these works. To the best of our knowledge this is mainly due to the fact that DARTS yields degenerate architectures in a wide range of search spaces. Works like [1] aim to deal with these problems and can be easily incorporated in FEATHERS as well. We already did this and performed some experiments with the more robust variant of DARTS and added the results.
> >
> > How does the improved version of DARTS compare in this setting? Does your method also suffer from the issues reported by Yang et al.?
> >
> > > Architectural parameters as used in DARTS can be interepreted as a probability that a certain operation defined in the search space is optimal for minimizing the loss. This implies - assuming DARTS yields the correct "distribution" over operations - an attacker can easily infer which operation(s) transform the data best s.t. the loss is optimized. Hence architectural parameters can guide attackers regarding "where to search" for private information in the network. This can be mitigated/avoided by applying DP on architecture parameters as well.
> >
> > I still fail to see how an attacker can extract any private information included in the data just by knowing that one operation performs better than another operation? For example, which potential private information could be revealed if I know that conv1x1 performs better than conv3x3 operation in a specific location of the network?
> >
> > > We agree that there are overlaps in the uncertainty bars. However, we emphasize that we solve a harder optimization problem than DARTS does since we optimize for additional hyperparamters. Nevertheless we still achieve better results than DARTS on average.
> >
> > It is correct that FEATHER solves a harder problem. But the problem remains, that uncertainty bars are too high to conclude that FEATHER actually significantly outperform DARTS. Just comparing the average ignores this uncertainty.
> >
> > > We do not provide exact convergence results of DARTS. To the best of our knowledge there is no rigor convergence analysis on DARTS.
> >
> > If there is no convergence analysis for DARTS, than it might be better to remove the sentence: “..FEATHERS’
> > convergence properties subsequently coincide with those of DARTS..” to avoid confusion.
> >
> > > Since the training round can possibly comprise an arbitrary number of epochs of local training, we could possibly train the (super-)net until convergence for each selected hyperparameter. Since this is computationally infeasible we only train for a few epochs locally (usually less than 5) to see whether the selected hyperparameter optimizes the loss well enough. This follows a similar rationale as early stopping methods which consider a few epochs of training enough to approximate the final model performance (see e.g. Li et al. Hyperband: A Novel Bandit-Based Approach to Hyperparameter Optimization. 2016).
> >
> > Hyperband (and most other multi-fidelity methods) subsequently increases the number of epochs for well-performing configuration simply because one epoch is often not sufficient to distinguish between top configurations.

---

> > > ### Author Response · Authors · 2023-08-29
> > > **Author Response**
> > >
> > > Thank you for participating on the discussion! We'd like to address your remaining concerns below:
> > >
> > > > How does the improved version of DARTS compare in this setting? Does your method also suffer from the issues reported by Yang et al.?
> > >
> > > We did not find a significant improvement in performance using DARTS with path dropout (the improved DARTS version as shown in [1]), see the following table:
> > >
> > > |      |    no label skew  |   label skew   |
> > > | ---- | ---- | ---- |
> > > |   FEATHERS + PDrop (5 Clients)   |   0.92 $\pm$ 0.02   |   0.91 $\pm$ 0.02   |
> > > |   FEATHERS + PDrop (10 Clients)  |   0.92 $\pm$ 0.02   |   0.90 $\pm$ 0.04   |
> > > |   FEATHERS + PDrop (100 Clients)  |   0.90 $\pm$ 0.03  |   0.90 $\pm$ 0.03   |
> > >
> > > We did not explicitly examine FEATHERS w.r.t. the issues reported in [2] However, we believe that the additional HO phases of FEATHERS can have a significant impact on the final performance as [2] showed that the optimization of other hyperparameters than the architecture has high impact in the DARTS evaluation pipeline. Since we apply HO during NAS and for evaluation, FEATHERS might find better suited architectures and might manage to find better hyperparameters to evaluate them. However, for this further research has to be done which exceeds the scope of our work.
> > >
> > > > I still fail to see how an attacker can extract any private information included in the data just by knowing that one operation performs better than another operation? For example, which potential private information could be revealed if I know that conv1x1 performs better than conv3x3 operation in a specific location of the network?
> > >
> > > The point we tried to make is the following: It's not that the architectural weight itself allows the attacker to extract private information. Rather it helps the attacker to "search for" private information in the right way. To stick with your example of conv1x1 vs conv3x3 operations, consider the following: If we know that conv1x1 found to work better than conv3x3 in the first layer, the attacker can conclude that the color-/channel-information seems to be important for the network's performance. Hence the attacker can try to identify neurons being activated for certain color-patterns and that way extract private information. In other words: An attacker can use such information to guide the extraction process.
> > >
> > > > It is correct that FEATHER solves a harder problem. But the problem remains, that uncertainty bars are too high to conclude that FEATHER actually significantly outperform DARTS. Just comparing the average ignores this uncertainty.
> > >
> > > We agree and have adjusted the wording to "FEATHERS reaches state of the art results" instead of stating that it outperforms SOTA methods like DARTS.
> > >
> > > > If there is no convergence analysis for DARTS, than it might be better to remove the sentence: “..FEATHERS’ convergence properties subsequently coincide with those of DARTS..” to avoid confusion.
> > >
> > > We think stating this point is crucial to show that FEATHERS has the same convergence properties as DARTS has (although DARTS' convergence properties are unknown this is correct). However, we see that this could lead to confusion and rephrased the sentence to "without stating DARTS' convergence properties explicitly, we can conclude that FEATHERS' convergence properties coincides with those of DARTS".
> > >
> > > > Hyperband (and most other multi-fidelity methods) subsequently increases the number of epochs for well-performing configuration simply because one epoch is often not sufficient to distinguish between top configurations.
> > >
> > > We agree on that, but note that there is no restriction on the number of epochs we use to train the supernet during HO. Please note that one communication round is not the same as one epoch of training: In one communication round the clients can perform an arbitrary number of epochs to train their local (super-)net, in our experiments we set the number of epochs to 3. We are aware that this might not be enough to distinguish between two configurations of hyperparameters reliably. However, reliably identifying the best hyperparameter configuration can only be achieved with a full training of the (super-)net in each communication round (or even multiple ones to account for stochasticity of SGD) which is infeasible and which also holds for methods like Hyperband.
> > >
> > > [1] Zela, Arber, et al. "Understanding and Robustifying Differentiable Architecture Search." International Conference on Learning Representations. 2019.
> > >
> > > [2] NAS evaluation is frustratingly hard Antoine Yang, Pedro M. Esperança, Fabio M. Carlucci. 2019.

---

> ### Author Response · Authors · 2023-08-21
> **Rebuttal (2/2)**
>
> > What's the rationale behind confining the optimization to discrete hyperparameters?
>
> The rationale was to show that a method performing both, hyperparameters and architecture, is feasible in FL settings. As mentioned in Section 5 we consider this to be a limitation of FEATHERS. Extending the hyperparameter optimization method of FEATHERS to one which is capable of handling continuous and discrete parameters is left for future work.
>
> > Why is a single round of training deemed a suitable approximation for optimal parameters w and architecture parameters a?
>
> Since the training round can possibly comprise an arbitrary number of epochs of local training, we could possibly train the (super-)net until convergence for each selected hyperparameter. Since this is computationally infeasible we only train for a few epochs locally (usually less than 5) to see whether the selected hyperparameter optimizes the loss well enough. This follows a similar rationale as early stopping methods which consider a few epochs of training enough to approximate the final model performance (see e.g. Li et al. _Hyperband: A Novel Bandit-Based Approach to Hyperparameter Optimization_. 2016).
>
> > In Equation 5, why is the difference taken here rather than the most recent observed validation error? What does w'/a' represent?
>
> We take the difference to estimate the **progress** a certain hyperparameter configuration achieved in terms of minimizing the loss. If a hyperparameter configuration worked well, the loss decreases, i.e. the difference is positive. If the opposite is the case, the difference gets very small or even negative (i.e. we diverged). Large positive differences (i.e. high decrease in loss) are rewarded afterwards while negative differences are "punished". $\mathbf{w}'$ and $\mathbf{a}'$ refer to the model-/architectural parameters obtained by locally training the supernet.
>
> > Why was the choice made not to jointly optimize hyperparameters and architectural parameters?
>
> Thank you for the interesting reference! In fact we do perform joint hyperparameter and architecture search. However, we formulate the optimization problem as a bi-level optimization problem which alternates between updating the architecture and other hyperparameters.
>
> > In the experiments, why is the skewed label experiments referred to as non-i.i.d.? How exactly are data points sampled?
>
> We referred to label skew as non-i.i.d. as the labels are not identically distributed over the clients. However, we find the naming suboptimal and changed it to "label-skew scenario". We sample the datapoints by defining a distribution of labels for each client. The default case is a uniform distribution (i.e. each client holds the same number of instances of each label). By skewing the distribution over labels, we enforce that some clients receive more instances of certain labels. For example, in a binary classification problem with 2 clients, we assign 75% of negative labels to client 1 and 25% to client 2. In turn, we assign 25% if positive labels to client 1 and 75% of positive labels to client 2.
>
> > How does FedEx work?
>
> FedEx also considers hyperparameter optimization over a discrete set of configurations. It defines a prior over configurations, samples one configuration in each round based on that distribution and updates the distribution based on the resulting loss returned by the clients, similarly as we do. However, they use an exponentiated gradient update which turned out to be very unstable for many problem instances such as image classification. For more information refer to _Khodak et al. Federated Hyperparameter Tuning: Challenges, Baselines, and Connections to Weight-Sharing. 2021._
>
> [1] Zela et al. Understanding and Robustifying Differentiable Architecture Search. 2019.

---

### Author Response · Authors · 2023-08-21
**Rebuttal**

We thank all reviewers for their thoughtful feedback as it has helped improving our paper.

We are also pleased that all reviewers appreciate our contribution.

Also we see that there is a high interest in our work proposing a novel method to optimize hyperparameters and neural architectures jointly.

We uploaded a revised version of the paper and highlighted requested changes using the following color coding:

magenta = Reviewer R8pX

violet = Reviewer fB8p

blue = Reviewer ocob

orange = General changes

Kind regards,

your authors.

---

### Decision · Action_Editors · 2023-10-16

**Recommendation:** Reject

**Comment:**

The reviewers found the effort to combine hyperparameter search with neural architectural search (NAS) in the FL setting interesting, and raised some concerns on the motivation of differential privacy (Reviewer ocob) as well as on its implementation and comparison to baselines (Reviewer fB8p). Overall, the current draft lacks some crucial details and some experimental results are perhaps a bit too good to be true (Reviewer R8pX). Given TMLR's requirement on correctness, we believe a major revision is needed and therefore would not recommend acceptance at this stage.

Below is a summary of the main concerns that I share with the reviewers:

-- The motivation on differential privacy in NAS and hyperparameter search is a bit weak. The authors' response on this point sounds speculative and lacks some rigorous evidence.

-- The DP guarantees and experiments are not convincing. The authors appear to simply equate "noise addition" with DP, without much care to the details. For instance, the authors claimed epsilon-DP throughout but added Gaussian noise in the algorithms, while it is a well-known fact that the Gaussian mechanism only achieves (epsilon,delta)-DP but not epsilon-DP. Reviewer R8pX asked about DP accounting, while the authors' response showed a concerning lack of understanding of this question. I also agree with Reviewer R8pX that Fig 4 is perhaps *too* good to be true, and warrants further investigation and explanation. There was no mentioning on how $\sigma_\theta$ and $C_\theta$ are chosen in Eq (11), how the batch size $B$ affects $epsilon$, and how clipping is done in Algorithm 4 and 5. There is no explanation on lines 1 and 5 of Alg 4 where one adds Gaussian noise to the objective function (instead of gradient) to achieve what DP guarantee (and how)? The update on $a$ (after equation (10)) does not appear to be a simple SGD update, and therefore does the DP analysis of Abadi et al. apply? The lack of these crucial details makes it impossible to verify the correctness of the authors' claims.

**Audience:**

Yes.

**Claims And Evidence:**

Partially. The results on differential privacy are not convincing and do not contain sufficient details for verification.

**Resubmission Of Major Revision:**

The authors may consider submitting a major revision at a later time.